# Recent Advances in Psoriasis Research; the Clue to Mysterious Relation to Gut Microbiome

**DOI:** 10.3390/ijms21072582

**Published:** 2020-04-08

**Authors:** Mayumi Komine

**Affiliations:** 1Department of Dermatology, Jichi Medical University, 3311-1 Yakushiji, Shimotsuke, Tochigi 329-0498, Japan; mkomine12@jichi.ac.jp; Tel.: +81-285-58-7360; 2Department of Biochemistry, Jichi Medical University, 3311-1 Yakushiji, Shimotsuke, Tochigi 329-0498, Japan

**Keywords:** psoriasis, tissue resident cells, innate lymphoid cells, regulatory T cells, Foxp3, gut microbiome, systemic inflammation

## Abstract

Psoriasis is a chronic inflammatory cutaneous disease, characterized by activated plasmacytoid dendritic cells, myeloid dendritic cells, Th17 cells, and hyperproliferating keratinocytes. Recent studies revealed skin-resident cells have pivotal roles in developing psoriatic skin lesions. The balance in effector T cells and regulatory T cells is disturbed, leading Foxp3-positive regulatory T cells to produce proinflammatory IL-17. Not only acquired but also innate immunity is important in psoriasis pathogenesis, especially in triggering the disease. Group 3 innate lymphoid cell are considered one of IL-17-producing cells in psoriasis. Short chain fatty acids produced by gut microbiota stabilize expression of Foxp3 in regulatory T cells, thereby stabilizing their function. The composition of gut microbiota influences the systemic inflammatory status, and associations been shown with diabetes mellitus, cardiovascular diseases, psychomotor diseases, and other systemic inflammatory disorders. Psoriasis has been shown to frequently comorbid with diabetes mellitus, cardiovascular diseases, psychomotor disease and obesity, and recent report suggested the similar abnormality in gut microbiota as the above comorbid diseases. However, the precise mechanism and relation between psoriasis pathogenesis and gut microbiota needs further investigation. This review introduces the recent advances in psoriasis research and tries to provide clues to solve the mysterious relation of psoriasis and gut microbiota.

## 1. Introduction

Psoriasis is a chronic inflammatory skin disease, with diverse disease severity and clinical phenotypes. Currently, no curative treatment exists, however, recent advances in therapeutics have made it possible to suppress the disease almost to disappearance. Progress in immunology and molecular biology has dramatically changed understanding of psoriasis pathogenesis, enabling development of novel treatment modalities.

In this review, several recent findings in psoriasis research are reviewed, to support further investigation.

## 2. Present Understanding of Psoriasis Pathogenesis

Plaque psoriasis is clinically defined as a cutaneous disease with multiple plaques of scaly erythematous lesions. The diagnosis is usually simple with macroscopic findings, but sometimes difficult to differentiate from eczematous, lichenoid, or other conditions. Microscopic investigation is sometimes helpful, but macroscopically difficult cases are also usually difficult to diagnose microscopically. The reason may be that there are heterogeneous clinical groups of psoriasis patients, suggesting that there are heterogenous pathogenetic groups in psoriasis patients. This is supported by the fact certain biologic drugs are efficient in some psoriasis patients but not in others. It has also been postulated that there is mixed pathogenesis of autoimmune and autoinflammatory conditions in psoriasis [1], and the balance of these conditions may differ among patients.

Krueger and Guttman [2] proposed a novel concept which positioned psoriasis and atopic dermatitis at the both ends of the same inflammatory disease spectrum: Psoriasis is located at pure Th17 inflammation, and atopic dermatitis is a combination of Th17, Th2, and Th1 inflammation.

Recent development of biologics for psoriasis treatment produced a novel skin condition, named “paradoxical psoriasis-like eruption” [3]. Paradoxical reactions have been noted for the first time in rheumatoid arthritis patients treated with anti-tumor necrosis factor (TNF) antibodies who developed psoriasis-like skin eruptions. Recently other immunological conditions provoked by targeted biological agents are included in paradoxical reactions, considered “class-effect of targeted biological agents”. The interaction between genetically predisposed conditions and targeted biological agents may result in the development of paradoxical reaction development. This itself is of scientific interest, in the context of understanding mechanisms of inflammatory diseases.

The present understanding of psoriasis pathogenesis is summarized in Figure 1. Dendritic cells, T lymphocytes, and keratinocytes are the three major cellular players. Genetic background predisposes patients susceptible to psoriasis: epithelial cells exposed to various stimuli such as bacteria, virus, ultraviolet light, or mechanical stress, leads to apoptosis or necrosis of these cells resulting in exposure of self nucleic acids to tissues [4]. Self DNA bound to LL-37, a part of antimicrobial peptide cathelicidine produced by keratinocytes (KC), stimulating plasmacytoid dendritic cells (pDC) to produce large amount of type I interferons. Simultaneously, self RNA bound to LL-37 stimulate myeloid dendritic cells (mDC) produce tumor necrosis factor alpha (TNFα) and inducible nitric oxide synthase (iNOS). These cytokines produced by DCs stimulate immature T cells to develop into inflammatory T cells, especially Th17 cells, producing interleukin (IL)-17 and IL-22, which develop psoriatic phenotype in KC. KCs produce antimicrobial peptides such as cathelicidine, beta defensine (BD), psoriasin and S100 proteins, chemokines e.g., CXCL1, 2, 8, 10, 11, and CCL20 which attract neutrophils and Th17 cells; and proinflammatory cytokines, such as TNFα, IL-1, and IL-17. These inflammatory reactions cause an inflammatory loop sustaining chronic psoriasis [4].

Recent findings also imply psoriasis inflammation’s systemic nature. Obesity’s genetic background is distinct from that of psoriasis, however, psoriasis is highly associated with obesity. One explanation is adipose tissue in obese persons produce inflammatory adipocytokines, such as leptin, resistin, and TNFα, leading genetically predisposed patients to develop psoriasis. Increased amount of inflammatory adipocytokines and decreased regulatory adipocytokines have been reported in psoriasis patients, which also correlate with disease severity. This adipocytokine imbalance accelerates insulin resistance and endothelial dysfunction, leading to atherosclerosis and cardiovascular events [5].

## 3. Tissue Resident Cells

Tissue resident cells have been identified and have focused attention on the possibility of disclosing one of the unsolved mysteries: why systemic inflammation of psoriasis chooses skin and articular as the most involved inflammatory sites.

Clark R. et al. [6] proposed resident memory T cells as pathogenic cells of fixed and recurrent skin lesions of psoriasis. Resident memory T cells are a recently identified T cell subset residing in epithelial barrier tissues such as skin, gut, lung, and reproductive tracts. They are highly protective against pathogens frequently encountered in each tissue. Boyman O et al. [7] in 2004 disclosed that normal appearing, uninvolved skin of psoriasis patients transplanted on immune deficient mice developed psoriatic lesions, indicating that there existed pathogenic skin resident immune cells in non-lesional skin of psoriasis patients. Subsequent studies revealed almost 20 billion T cells reside in healthy human skin, twice as many T cells in the entire blood volume. These skin resident T cells express CD45RO, CLA, and CCR4, and strong effector functions with various T cell receptor (TCR) repertoires. Large numbers of antigenically active tissue resident T cells have subsequently been disclosed in the gastrointestinal tract, lung, reproductive tract, peritoneum, and bone marrow [8,9,10,11,12,13].

Clinical trials of anti-E-selectin which completely block the migration of circulating T cells from blood to skin on psoriasis patients revealed its ineffectiveness, suggesting the clue to Tissue resident cell involvement in psoriasis [14]. Another study demonstrating transplanted normal-appearing uninvolved skin of psoriasis patients on immunodeficient mice developed psoriasis lesions suggested psoriasis patients’ normal-appearing skin contains pathogenic cells which can develop psoriasis lesions without the circulating blood cells [7]. Subsequent studies revealed CD4-positive T cells with IL-22 and CD8-positive T cells producing IL-17 remained resident in previously involved psoriasis lesions after complete clinical resolution from a variety of treatments [15,16].

Gallais serezal I et al. [17] proposed a skewed pool of resident T cells in psoriasis patients trigger non-lesional skin of psoriasis patients to develop psoriasis lesion. They collected never lesional psoriasis (NLP) skin from mild psoriasis patients instead of from severe psoriasis, because in severe psoriasis, patients’ uninvolved skin shows psoriatic gene expression profiles. These NLP with normal gene expression profiles exhibited IL-17- and IFN-γ-producing T cell accumulation after stimulation, which triggers keratinocytes to produce IFNα introducing an alternative source of Type I interferons in initiation of psoriasis. Their analysis revealed skewed T cell population in NLP with CD8-positive CD49a-negative T cells, and CCR6-positive T cells producing IL-22, IL-17, and IFNγ. NLP keratinocytes produce increased amounts of CCL20 upon stimulation with *Candida albicans* or mannan suggesting the role of KC, weakly stimulated by resident fungi, in accumulation of IL-17 and IL-22 producing CCR6-positive resident T cells in NLP skin of psoriasis patients. Resident T cells are insufficient to develop psoriasis lesions, but upon stimulation with various stimuli, such as mechanical stress, viral infection, and bacterial infection, these skewed population of resident memory T cells produce inflammatory cytokines, such as IL-17, IL-22, and IFNγ, which initiate psoriasis inflammation (Figure 2). Their study also demonstrated for the first time the role of KC in production of IFNα in psoriasis pathogenesis.

## 4. Innate Lymphoid Cells

Innate lymphoid cells (ILCs) are recently identified innate immune cells. They were found among lineage-negative cell populations not expressing any mature cell markers, such as Mac-1 for myeloid cells, CD4 and CD8 for T cells, CD19 for B cells, or Ter-119 for erythrocytes. They have a high potency for cytokine production without specific antigen stimulation [18]. Psoriasis has been considered an “autoimmune” disease; however, no specific antigens have been identified so far. The innate immune system is a relatively primitive immune system, directly activated by components of virus, bacteria, fungi, or self-DNAs and RNAs, without specific antigens. Many cell types were included in innate immune cells, such as neutrophils, mast cells, and natural killer cells as well as innate lymphoid cells. It is not surprising that these innate immune cells take part in psoriasis pathogenesis.

ILCs have been subdivided into three groups: ILC1, ILC2, and ILC3. ILC1 (group 1 ILC) characteristically produces IFN gamma, perforin, and granzyme, while ILC2 (group 2 ILC) produces IL-5, IL-13, and IL-4, with nuclear expression of GATA3. ILC3 (group 3 ILC) expresses IL-7Rα, matures with IL-7 and IL-23, and produces IL-17, IL-22, and lymphotoxin with nuclear expression of RORγt [18,19]. ILC3 in humans is further divided into three subsets on the basis of expression of natural cytotoxicity receptors: NKp44, NKp46, and NKp30. NKp44-positive ILC3 produces IL-22 and is dependent on aryl hydrocarbon receptor (AhR). NKp44-negative ILC3 produces IL-17A following stimulation, but NKp44-negative ILC3s have plasticity, and they are able to develop into NKp44-positive ILC3s or into ILC1s with IFNγ production [20]. In inflammatory bowel diseases, ILC3s have been shown to produce IL-17 in the gut [21]. Therefore, the world of ILCs looks as if it were the parallel world of helper T cells (Figure 3).

Soare A et al. [22] reported increased numbers of ILC3 exist in the circulating blood of psoriatic arthritis patients, and ILC3/ILC2 ratios correlated well with disease severity. ILC3 also increased in the lesional and non-lesional skin of psoriasis patients. Keren A et al. [23] revealed that injection of NKp44-positive ILC3 without T cells was able to develop psoriatic skin eruption in SCID mice with implantation of healthy human skin.

These results indicate that ILC3 may have a pathogenic role in psoriasis, which compensates the absence of specific self-antigen.

## 5. Regulatory T Cells

Regulatory T cells (Treg) have immune suppressive function to suppress excess immunity against a diverse range of antigens, such as self-antigens, commensal bacteria-derived antigens, and environmental allergens. Many autoimmune and inflammatory diseases, such as systemic erythematosus, inflammatory bowel diseases, and rheumatoid arthritis, show decreased Treg numbers and function. Foxp3 (Forkhead box P3) is a transcription factor playing a crucial role in development, maintenance and function of Tregs. Deficiency of Foxp3 results in lack of Tregs, and causes severe systemic inflammatory diseases characterized by autoimmunity, colitis, and allergies.

Tregs usually develop in the thymus from CD4−, CD8−, double negative thymocytes, called thymus-derived regulatory T cells (tTregs) or natural occurring Tregs (nTregs). Expression of Foxp3 are strongly induced through T cell receptor (TCR) signals after the recognition of self-antigen-MHC complex present on antigen presenting cells (APCs). tTregs are believed to comprise most of the systemic Treg population. A second subset of Tregs is induced in the peripheral tissues from CD4-positive naïve T cells by stimulation with cytokine combination, such as TGFβ and IL-2. This type of Tregs is called induced Tregs (iTregs) or peripherally induced Tregs (pTregs). iTregs usually represent Tregs induced in vitro, while pTregs indicates Tregs developed from naïve T cells in vivo.

tTregs matured in thymus have constitutive and stable expression of Foxp3, supported by the binding of several transcription factors to its corresponding promoter and enhancer regions. Several enhancer regions are identified, designated as conserved non-coding sequences (CNS) 1, 2, and 3, and CNS0 has recently been identified upstream of CNS1 [24].

Promoter region and each CNS binds specific transcription factors, which regulate the expression of Foxp3.

The most fundamental transcription factor for Foxp3 expression may be the recently identified Satb1, which binds to CNS0 inducing both transcriptional and epigenetic regulation. Satb1 binds to CNS0 at the beginning of Treg commitment and serves as a pioneering element involved in the transcriptional regulation sequence in Treg development. Nr4a is a transcription factor which binds to CNS2 and the promoter region of Foxp3, necessary for both pTerg and tTreg development. TCR activation induces Nr4a expression, which is essential for pTreg and tTreg development and maintenance after TCR activation. IL-2 and TGFβ are well-known inducers of Tregs, and IL-2 is essential in inducing both tTregs and pTregs, while TGFβ is needed in pTreg development. STAT5 is activated downstream of IL-2R, and STAT5 response elements exist in CNS2 and Foxp3 promoter region. CNS1 contains binding sites for Smads, NFAT, AP-1, and retinoic acid receptor (RAR). Smad2 and Smad3 are redundantly essential for development of iTregs/pTregs in the downstream of TGFβ. CNS2 region contains biding sites for multiple transcription factors, such as STAT5, NFAT, Runx1/Cbfβ, CREB, and Foxp3. This enhancer is very important in maintaining its own expression under inflammatory circumstances where Tregs are subjected to inflammatory cytokines and stronger TCR stimulation. CNS2 locus has multiple CpG sites making this region susceptible for methylation/demethylation status. In tTregs, CNS2 is fully demethylated: full sets of transcription factors bind to this locus enabling stable expression of Foxp3, and stable phenotype of tTregs. In pTregs, CNS2 locus is also demethylated, but with a slightly lower stability compared to tTregs. In iTregs, CNS2 locus is rarely demethylated, which makes this subset very unstable. This CNS2 locus is the major Treg-specific demethylated resion (TSDR) whose demethylated status confers the stability of Foxp3 expression, and the function of Tregs [25] (Figure 4).

The inducers and stabilizers of Foxp3 expression are summarized in Table 1.

Retinoic acids and vitamin D3 are ligands for their specific nuclear receptors, retinoic acid receptor (RAR), retinoid X receptor (RXR), and vitamin D receptor (VDR). Retinoids binds to RARs and/or RXRs, which make homo or heterodimers to bind to their binding sequences in regulatory region in the target genes. Butyrate induces retinoic acid production in gut dendritic cells, resulting in induction and stabilization of Foxp3 expression in regulatory T cells.

Active form of vitamin D3, generated through the enzymes produced in skin or in liver and kidney, binds to VDR, which makes heterodimers with RXR and binds to response element usually resides in distal area of target genes and exert its functions. Ultraviolet (UV) B and antimicrobial peptides, such as cathelicidin and S100 proteins, are inducers of active vitamin D3 production in epidermis. Active vitamin D3 systemically distribute through blood flow binding to vitamin D binding protein (DBP), enter target cells through binding to heat shock protein (HSP) 70 in cytoplasm, and finally transported to the nucleus binding to VDR [26]. VDR binds to its binding site in enhancer regions of Foxp3 gene, resulting in induction and stabilization of Foxp3 expression. Alternative receptors other than VDR have been reported, such as retinoid orphan receptor (ROR)α, RORγ [27], and aryl hydrocarbon receptor (AhR) [28]. By binding to these alternative receptors, vitamin D3 derivatives have been reported to suppress inflammatory reactions, such as Th17-type inflammation, through suppressing the transactivation function of these nuclear receptors [27].

pTregs are induced by CD103-positive dendritic cells (DCs) in mesenteric lymph nodes dependent on TGFβ and retinoic acid. Cutaneous CD103-positive DCs have also been reported to induce pTregs [29]. Recent report indicated that DCs in cervical lymph nodes induce Tregs, but are not CD103-positive DCs [30,31].

Recently, Gagliani et al. [32] reported intestinal Th17 can lose the ability to produce IL-17 and behave like regulatory T cells resembling CD4+Foxp3- Type 1 Tregs (Tr1). This functional reprogramming is irreversible, and these transdifferentiated Tr1 cells show anti-inflammatory properties and suppress Th17-mediated colitis.

Psoriasis patients have a decreased number of regulatory T cells with disturbed function. Several reports indicated that successful treatment of psoriasis restored the function and number of Tregs in peripheral blood [33,34]. Bovenschen [35] reported that Foxp3-positive regulatory T cells are easily converted into Foxp3-low, IL-17 producing, and RORγt-expressing cells. The plasticity of Treg/Th17 cells makes pathophysiology of the disease more complex (Figure 5).

## 6. Microbiome

Advancement in computer science and next generation sequencing (NGS) technique has enabled analysis of huge amounts of genetic data at once, revealing composition of microorganisms without culture. High throughput 16S rRNA sequencing of gut microbial populations or total genome sequencing and mass-spectrometry-based metabolomic analysis can characterize gut microbiome and metabolites. Gut microbiota is a dense and diverse microbial community composed of more than 100 trillion cells and 5 million genes over 3-fold and 100-fold more than host cells and genes. Gut microbiota is composed of thousands of species. However, the majority belong to six bacterial phyla: *Actinobacteria*, *Bacteroides*, *Firmicutes*, *Fusobacteria*, *Proteobacteria*, and *Verrucomicrobia*. Fungi, Archaea, protozoa, and viruses are also included in gut microbiota. The composition of microbiota is highly influenced by environmental factors such as food, drug, and hygiene conditions, and also dependent on age and genetic background. Microbiota functions as natural digestive organs, as they digest plant polysaccharides indigestible by the host, can biosynthesize essential amino acids and vitamins, and also detoxicate hazardous materials. Recent studies revealed they also have important roles in immune system development and resistance to pathogens. Biotransformation of drugs can occur through gut microbiota altering the effects of drugs expected in vitro studies or catalyzing them to cause undesirable effects. Human genome analysis has revealed genetic susceptibility to certain diseases, and analysis of microbiota as a “second genome” would reveal more important information for disease susceptibility and drug metabolism. Further studies on gut microbiota are needed to understand the precise mechanisms in disease susceptibility variance and drug effects and side effects profiles between individuals. Such research could enhance development of precision medicine [36].

Recent studies linked cardiac disease, insulin resistance, and metabolic syndrome to gut microbiota [37,38]. Several reports supported the contribution of gut microbiota in the production of TMA (trimethylamine), the precursor of TMAO (trimethylamine N-oxide), which is a known proatherogenic molecule independent of traditional cardiovascular disease (CVD) risk factors. TMAO is involved in host cholesterol metabolism and activates macrophages leading to increased risk of CVD, myocardial infarction, and stroke. Higher TMAO producers had more *Firmicutes* than *Bacteroides* within the stool. TMAO has also been suggested to be a candidate molecule of developing type 2 diabetes mellitus (DM). Dietary supplementation with TMAO in mice resulted in impaired glucose tolerance and adipose tissue inflammation promotion. Crasiun and Balskus reported that *cutC* gene expression by bacteria such as *Disulfovibrio* can cause increased conversion of choline to TMA. Compared to healthy controls, phyla *Bacteroides* and *Proteobacteria* were reduced and phyla *Firmicutes* and *Fusobacteria* were increased in coronary artery diseases (CAD) patients [39]. Similar results were reported revealing decrease in phylum *Bacteroides* and increase in phylum *Firmicutes* in CAD patients [40]. Sanchez-Alcoholado et al. [36] compared CAD patients with type 2 DM to CAD patients without type 2 DM and revealed these two groups had different composition of gut microbiota. CAD patients with type 2 DM had decreased phylum *Bacteroides* and increased phyla *Firmicutes* and *Proteobacteria*. They also revealed plasma TMAO levels were significantly higher in CAD with DM patients compared to CAD without DM patients, which correlated with the increase in phyla *Enterobacteriaceae* and *Desulfovibrio* and a decrease in phylum *Faecalibacterium*.

Obesity is related to gut dysbiosis and increased gut permeability, resulting in abnormal translocation of bacteria and bacterial component in blood circulation. The increase in gut barrier permeability caused increase in blood lipopolysaccharide (LPS) levels, resulting in systemic low-grade inflammation and metabolic disease including type 2 diabetes [41]. Glucagon-like protein (GLP) 2 is produced by L cells in the gut, which is regulated by the host nutritional status, and increases nutrient absorption. It induces gut epithelial cell proliferation and expression of tight junction proteins. It also regulates production of antimicrobial peptides by Paneth cells. Gut microbiota in lean individuals have been reported to induce endogenous GLP2 production, resulting in improvement in gut barrier function, while that of obese patients suppresses GLP2 production leading to gut barrier impairment and bacterial translocation in blood [42].

Zonulin is a protein modulating permeability of tight junctions in the digestive tract. It was originally discovered in patients with Celiac disease and type 1 diabetes. Zonulin binds to its receptor and activates the pathway of tight junction openings resulting in increased gut permeability. Increased permeability in gut epithelial tissues leads to increased passage of antigens, and triggers autoimmunity in susceptible individuals. Recent study reported gliadin binds to CXCR3 triggering the release of zonulin. Zonulin activates PKC and PLC resulting in actin poymelization and rearrangement of cytoskeleton inducing loosening of tight junctions in a reversible manner. Increased serum level and increased excretion in the feces of zonulin have been reported in patients with gut barrier disruption, such as patients with Celiac disease, as well as type-1 diabetes, obesity, and non-alcoholic fatty liver disease [43].

Activated effector cells are anabolic, consuming glucose as their carbon source, and utilizing glycolysis to obtain ATP. Memory and regulatory cells are catabolic, utilizing fatty acids, amino acids, as well as glucose for their energy source, and utilize oxidative phosphorylation to obtain ATP. Key molecules promoting the glycolytic and lipogenic pathway are mammalian target of Rapamycin (mTOR) and adenosine monophosphate-activated kinase (AMPK). AMPK and mTOR are energy sensors regulated by nutrients availability. Th17 cells depend on glycolytic-lipogenic pathway and fatty acid synthesis for their development, while Tregs depend on oxidative phosphorylation and consume exogenous fatty acids. HIF1α is the transcription factor which upregulates glycolytic pathway in Th17 cells binding to RORγt promoter and enhancing its expression, while also suppressing Foxp3 expression. It promotes differentiation of naïve T cells towards Th17 cells and inhibits differentiation into Treg cells under normoxic and hypoxic conditions. Inflammation sites are usually in hypoxic condition and show increased extracellular ATP concentration, which induces HIF1αactivation required for Th17 cell development, and AhR inactivation needed in Tr1 metabolism. Thus, metabolic factors have immune-modifying ability by skewing Th17/Treg balance resulting in skewed balance in inflammation or immune tolerance [44].

Gut microbiota profoundly affect T cell differentiation and response to immune stimuli. Segmented filamentous bacteria (SFB), a *Clostridia*-related species found mainly in rodents, specifically induces Th17 cells in the small intestine and other sites in autoimmune condition. SFB colonization is usually beneficial because it attenuates bacteria-induced colitis, while it also induces colitis in genetically susceptible mice strains. The abundance of SFB and the gut barrier function is regulated by IL-23R/IL-22 pathway. The disruption of gut barrier resulting in systemic distribution of bacteria or their components induces IL-23 pathway, initiating barrier repair, and Th17 responses in order to neutralize invading microbes. IL-23 induced by SFB stimulates production of IL-22 from ILC3 causing serum amyloid A production by epithelial cells [45] (Figure 6).

Psoriasis is considered a systemic and chronic inflammatory disease, involving skin, as well as the cardiovascular system, insulin homeostasis, psychomotor systems, and lipid metabolism. A recent study has detected bacterial DNA in blood in active psoriasis patients (bacterial translocation; BT), and those with bacterial DNA-positive psoriasis patients showed higher serum inflammatory cytokine levels, longer duration of disease and younger onset of disease. They suggested that psoriasis eruption may be related to bacterial DNA in blood originating in the gut [46]. Codoner et al. revealed phylum *Bacteroides* decreased and phylum *Faecalibacterium* increased in psoriasis patients. They compared the microbiome between BT-positive patients and BT-negative patients and did not find specific phyla of microbiome. They speculated that dysbiosis in psoriasis patient guts caused insufficient gut barrier permeability resulting in bacterial translocation into the blood stream [47]. Hidalgo-Cantabrana et al. reported recently the increase in *Bacteroides* and decrease in *Firmicutes* in psoriasis patients compared to healthy controls [48].

*Bacteroides* can produce short chain fatty acids (SCFA) such as propionates and butyrates. These SCFA bind to the G-protein-coupled receptor, GPR41 and GPR43, and induce GLP-1 and GLP-2 production in L cells, which influence the energy metabolism and improve gut barrier function [49]. *Akkermansia muciniphila* has also been suggested to produce SCFA to improve energy metabolism, which are decreased in obese patients compared to healthy controls. SCFA are also involved in the stabilization of Foxp3 expression as discussed above, resulting in improvement of regulatory T cell function. GPR41 and GPR43, the receptors for SCFA, are expressed on the surface of regulatory T cells distributed in the gut. The number of Tregs increased in mice supplemented with SCFA. Experimentally induced autoimmune encephalomyelitis was suppressed in mice supplemented with SCFA [49].

Scher et al. [50] reported gut microbiota composition was different in psoriatic arthritis patients and psoriasis without arthritis patients. They found that phylum *Actinobacteria* was significantly decreased in psoriatic arthritis patients. *Actinobacteria* phylum includes *Bifidobacterium* species, of which supplementation lowered the serum levels of CRP and TNFαa in psoriasis patients.

Another approach altering the gut microbial effect is to influence the pathway the microbiome uses. Farnesoid X receptor (FXR) is the essential nuclear receptor whose antagonist can inhibit high-fat diet-induced obesity in mice. FXR is involved in the metabolism of bile acid, lipid, and glucose. Intestine-specific FXR knockout mice were resistant to high-fat diet-induced obesity. However, contradictory results have been reported: FXR-null mice showed increased glucose intolerance on a chow diet, and FXR-agonist was effective in improving insulin sensitivity in genetically obese mice. Other studies presented that FXR-null mice showed increased glucose tolerance and FXR-agonist exacerbated insulin resistance and lipid metabolism in obese mice models. Li et al. speculated this discrepancy is due to differences in gut microbiota composition and the distinct roles of liver FXR and intestine FXR. Their study provided evidence that FXR inhibition in the intestine by antioxidant “tempol” or by genetic ablation of FXR is effective in suppressing diet-induced obesity and insulin resistance [51,52]. Other molecules in the context of microbiome signaling are peroxisome proliferator activated receptor (PPAR)s. Gut microbiota influence the expression of PPARs, and PPARs transduce signals to induce or suppress many molecules involved in inflammation, obesity, and insulin resistance [53].

## 7. Discussion and Conclusions

Many novel findings revealed the importance of innate immunity in psoriasis pathogenesis. The roles of skin resident cells and Tregs have been drawing attention in psoriasis pathogenesis. Epidemiological study revealed the relation between psoriasis and comorbid diseases such as cardiovascular diseases, diabetes mellitus and obesity, and the gut dysbiosis in psoriasis patients has recently been demonstrated. However, the underling mechanism and pathogenic roles of gut microbiome has to be clarified further. This review tried to hunt the findings in basic science and in metabolic diseases, and to combine them to be applied to future psoriasis research.

It is not clear whether dysbiosis in gut microbiota in psoriasis patients has the pathogenic roles or just the result of systemic inflammation in psoriasis. However, the recent study on gut microbiota and the immune regulatory mechanism suggests that dysbiosis in gut microbiota by inducing disfunction of Tregs and activation of Th17 and ILC3 could lead genetically susceptible individuals to develop psoriasis. It may also cause obesity, diabetes mellitus, and cardiovascular diseases, among individuals with distinct genetic backgrounds. The increase in number of psoriasis patients in Japan with westernization of life could be explained by the change of diet, resulting in a change in microbiota. Decrease of dietary fibers may reduce SCFA-producing phyla in gut microbiota, resulting in activation of gut dendritic cells to produce IL-23, which in turn may stimulate unstable Foxp3-positive Tregs to differentiate into IL-17-producing cells. Psoriasis has been recognized as a multifactorial genetic disease, not only depending on genetic background, but also on environmental factors including lifestyle habits, which could also be well-explained.

Skewed balance of skin resident cells may be generated by activation of epidermal keratinocytes producing slightly higher amount of CCL20 by commensal microbe stimulation in genetically susceptible individuals, which could be enhanced under the environment of disturbed Treg function and increased Th17 cytokines in individuals with dysbiosis.

In this review, the gut–skin axis is the main target of discussion, thus skin-specific cells, such as epidermal keratinocytes and dermal fibroblasts, are not discussed. The nuclear receptors such as RARs, VDR, and RORs are expressed not only in immune cells, but also in epidermal keratinocytes [26], making the epidermal keratinocytes another target of immune regulation. Skin has its own microbiome, through influencing the homeostasis of epidermis, it regulates the immune status of the whole body. The discussion on these issues have been left at another opportunity.

Systemic inflammation in psoriasis patients has led researchers to investigate the relation of psoriasis to metabolic diseases and has significantly deepened recent psoriasis investigation. A systemic approach to treat psoriasis has long been attempted with etretinate and cyclosporine, and recent biologics have shown remarkable therapeutic effects. Knowledge in skin resident cells are preferential to topical treatment, however, accumulating evidence in the gut–skin axis [54] would favor a nutritional approach or signal inhibition to change microbiome composition. Therefore, increasing therapeutic options may be available for future psoriasis treatment.

## 8. Figures, Tables, and Schemes

External immune triggers, LL37 bound to self-DNA or RNA activate plasmacytoid and myeloid dendritic cells (pDC and mDC). Activated plasmacytoid cells produce type 1 interferons, and myeloid dendritic cells produce TNFα, iNOS, and IL-23, which activate Th17 cells to produce IL-17 and IL-22. IL-17 and IL-22 induce proliferation, production of cytokines, chemokines and antimicrobial peptides from epidermal keratinocytes, thus resulting in psoriatic phenotype.

Weak stimulation by commensal bacteria and fungi causes sparse production of CCL20 from the epidermal keratinocytes, resulting in skewed distribution of CCR6-positive, IL-17-producing cells in the never-lesional skin of psoriasis patients (NLP) over IFNγ-producing cells, which prepare conditions for psoriasis eruption when appropriate insults, such as bacterial or viral infection, or trauma occurs.

Innate lymphoid cells are divided into three groups; group 1, 2, and 3. Function and essential nuclear factors seem similar to those of corresponding T helper cells.

Promoter lesion of FOXP3 gene contains major Treg-specific demethylated resion (TSDR), which binds transcription factors, such as FOXP3, CREB, Ets1, and STAT5 constitutively in tTregs, but just transiently in pTregs. TSDR in pTregs are partially methylated, which makes it difficult to bind transcription factors needed for the stable expression of FOXP3.

tTregs differentiated in thymus stably express FOXP3 and are involved in immune tolerance. Peripherally induced Tregs and Tregs with unstable FOXP3 expression can lose FOXP3 expression and become inflammatory cells, which may contribute to autoimmunity.

Segmented filamentous bacteria (SFB) colonization allow them to closely contact to intestinal epithelium, which induce cytokines, such as serum amyloid A (SAA), and activate Th17. SAA also induces IL-23 production from dendritic cells, which indirectly activate Th17 and ILC3 to produce pro-inflammatory IL-17, and IL-22. IL-22 stimulate intestinal epithelial cells to produce SAA.

*Clostridia* is one of the commensal bacteria in the gut, which ferment dietary fibers and produce short chain fatty acids; butyrates. SCFA suppresses dendritic cell production of inflammatory cytokines, such as IL-6, and also induces retinoic acid (RA) production from dendritic cells, which induces and stabilizes FOXP3 expression in naïve Tcells. SCFA also directly act on naïve T cells to express FOXP3.

There are different strains of *Bacteroides fragilis*, one of which produces polysaccharide A (PSA) and induce IL-10 production from Treg cells, thereby inducing immune suppression. The other strain produces B. fragilis toxin (BFA), which causes impairment of tight junctions and disruption of intestinal barrier function. Recognition of microbial products from disseminated bacteria by microbe associated molecular patterns stimulates IL-23 production from monocytes and stimulates Th17 cells to produce IL-17 and exacerbates inflammation.

## Figures and Tables

**Figure 1 ijms-21-02582-f001:**
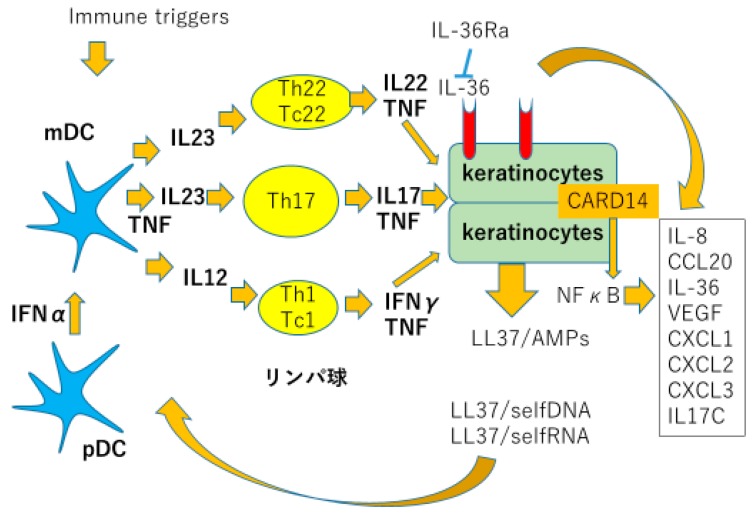
Pathogenesis of psoriasis. Modified from Lowes M.A. et al. Trends in Immunol 2016 [4].

**Figure 2 ijms-21-02582-f002:**
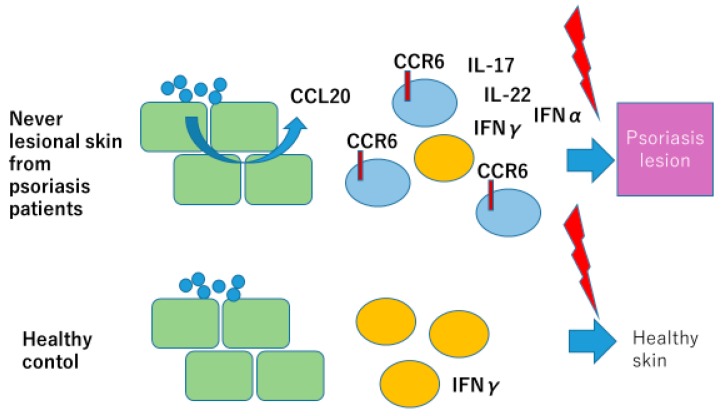
Skewed balance of tissue resident immune cells induces psoriasis in never-lesional skin of psoriasis patients. Modified from Gallais Sérézal I et al. J Allergy Clin Immunol 2018 [17].

**Figure 3 ijms-21-02582-f003:**
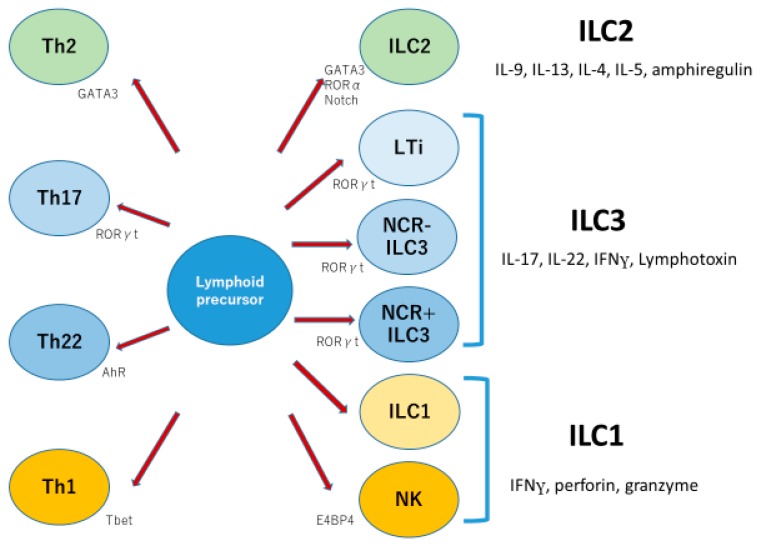
Three groups of innate lymphoid cells parallel three types of T helper cells. Modified from Walker JA et al. Nat Rev Immunol 2013 [18].

**Figure 4 ijms-21-02582-f004:**
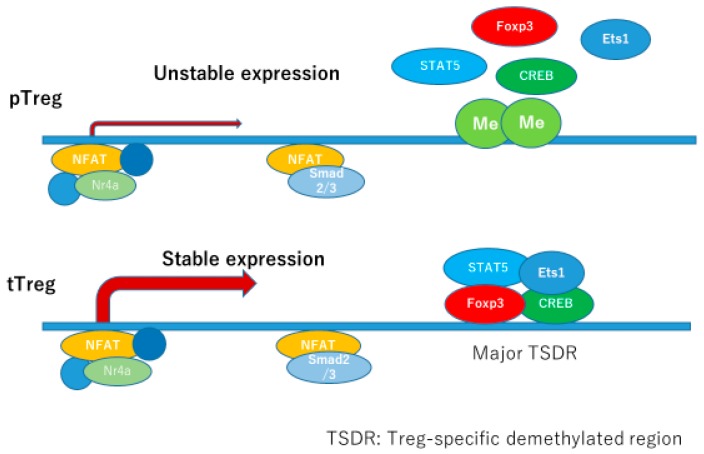
Expression of FOXP3 in pTreg is unstable. Modified from Kanamori M et al. Trends in Immunol 2016, and Iizuka-Koga et al. J Autoimmun 2017 [24,25].

**Figure 5 ijms-21-02582-f005:**
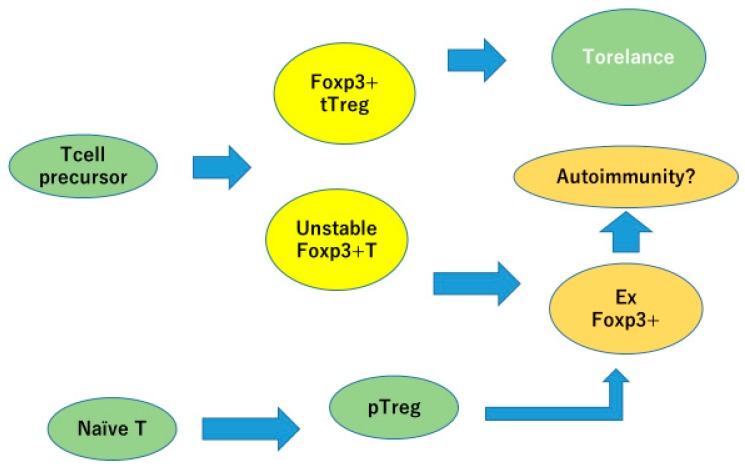
Foxp3-positive Tregs, but with unstable expression, would easily lose FOXP3 expression and become inflammatory. Modified from Iizuka-Koga M et al. J Autoimmun 2017 [25].

**Figure 6 ijms-21-02582-f006:**
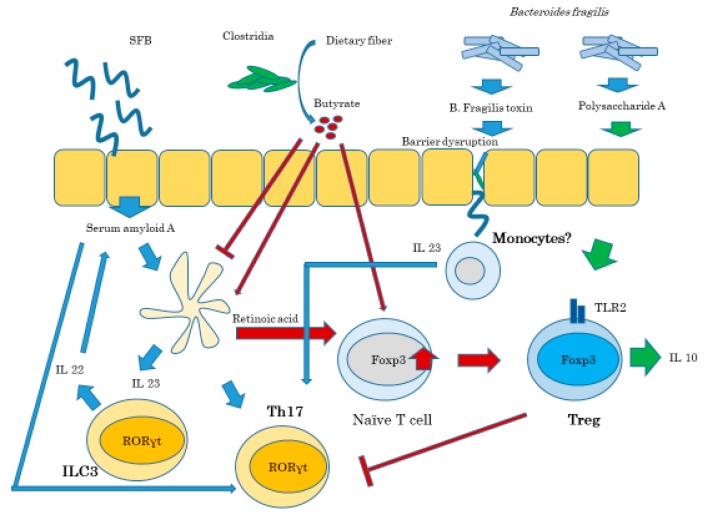
Gut microbiota regulates the balance of Th17 vs. Tregs. Modified from Omenetti S and Pizzaro TT. Frontiers Immuno 2015 [43].

**Table 1 ijms-21-02582-t001:** Factors which induce or stabilize Foxp3 expression.

Factors	Mechanism
Retinoic acids	Binding to *Foxp3* enhancer CNS1 through RAR
Progesterone	Suppression of mTOR
Vitamine D3	Binding to *Foxp3* enhancer CNS1 through VDR
Short chain fatty acids	Activation of GPR43
Butyrates	Inhibition of HDAC
Vitamine C	Activation of TET enzymes
Hydrogen sulfide	Induction of TET1 and TET2
Rapamycin	Inhibition of mTOR
JAK1 inhibitor	Suppression of Th17
AhR ligands	Suppression of Th17

Modified from Kanamori M. et al. Trends in Immunol 2016 [24].

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
