# Peer review of "Recent Advances in Psoriasis Research; the Clue to Mysterious Relation to Gut Microbiome"

_ijms, 2020, doi:10.3390/ijms21072582_

Round 1

Reviewer 1 Report

It is a very well written article. 

Author Response

Thank you very much for taking your time to review my manuscript. 

I have asked native English speaker to correct my manuscript.

Thank you again for your comment.

Reviewer 2 Report

The review of Mayumi Komine is about the relation of gut microbiome to psoriasis. The topic sounds very interesting, but I feel that the manuscript does not really shows any link of gut microbiome to psoriasis.

In current form, the review rather handles gut microbiome in other chronic inflammatory diseases. Although the title and abstract promises to find the putative link of gut microbiome to psoriasis, information on these aspects is almost entirely missing from the manuscript.

In the manuscript, there are extensive descriptions about innate lymphoid cells, T-reg cells and gut microbiome etc., but regarding these, little-to-no-information is presented about their link to psoriasis. Also, in the description of psoriasis, the role of keratinocytes is underrepresented, and links of the immune responses of keratinocytes to psoriasis (e.g. inflammasome activation, IL-36 production etc.) is missing.

Probably, there is so far little information in the literature about gut microbiome and its relation to psoriasis, but I still think, if the aim of the manuscript is to provide a link between gut microbiome and psoriasis, then a stronger focus should be set on psoriasis. It would be also interesting to show the links of skin microbiome and psoriasis parallel to the links of gut microbiome and psoriasis. The paper might be converted into a „concept paper” and describe the possible links of gut microbiome and psoriasis in more detail, might pinpointing the putative pathogenic aspects of gut microbiome to psoriasis.

Based on these, I do not think, that in present form the manuscript would fulfill its promise to show the links of gut microbiome to psoriasis, thus, I suggest an extensive revision.

Author Response

To reviewer 2

Thank you very much for your valuable comments.

As you mentioned, there has been little reports regarding psoriasis and gut microbiome. I tried to present reported information on basic research and area of metabolic syndrome, regulatory T cells, innate lymphoid cells, trying to combine them, which could be applied to future psoriasis research.

I added some descriptions to abstract and conclusions, in order to be in line with the content of the manuscript, trying to present conceptual proposal.

Round 2

Reviewer 2 Report

The author made changes to the abstract and discussion, but I still feel that the focus of the manuscript is not psoriasis, but the comprehensive listing of the knowledge on immune cell types. I think these parts, not related to psoriasis, shoud be shortened, to have a clearer overview on psoriasis, and not on unrelated information. I would suggest to rethink the concept and delete parts which are not necesarry to understand the psoriasis-link of gut-micorbiote.

As I suggested before, the psoriasis description should be amended with the role of keratinocytes, and links of the immune responses of keratinocytes to psoriasis (e.g. inflammasome activation, IL-36 production etc.).

My other note is that the figure captions are partially in Japanese. This has to be corrected, before acceptance.

Thus I think, the manuscript has to undergo extensive amendments to be acceptable.

Author Response

Thank you very much for your important comments.

I reconsider what you would like to tell in your comments, and I really understand that the role of keratinocytes in the pathogenesis of psoriasis research is very important as you suggested, and that is what I have been involved in. However, I think that the manuscript I submitted in its present form is what I want to describe in this review. In this review, I would like to focus on the relation between immune status of the host and the gut microbiome, that is why the discussion on immune cells predominates over psoriasis itself. I would like to convey the importance of microbiome to host immune system and the metabolic system, which could influence on the pathophysiology of psoriasis, while the latter part has not been extensively investigated, resulting in less description. The molecular mechanism of Foxp3 expression should be described, and the role of ILC3 should be demonstrated in order to fully understand the relation to microbiome. The pathogenesis of psoriasis has been discussed everywhere and lot of people already know the importance of IL-36 production and keratinocyte activation. That is why I omitted this part.

I hope you would kindly understand what I intended to describe in this manuscript.